# Effect of custom-made and prefabricated foot orthoses on kinematic parameters during an intense prolonged run

**Marina Gil-Calvo**[1☺], **Irene Jimenez-Perez**[1,2☺]*, **Jose Ignacio Priego-Quesada**[1,2☺], **Ángel G. Lucas-Cuevas**[1☺], **Pedro Pérez-Soriano**[1☺]

**1** Department of Physical Education and Sports, Research Group in Sport Biomechanics (GIBD), University of Valencia, Valencia, Spain, **2** Department of Physiology, Research Group in Medical Physics (GIFIME), University of Valencia, Valencia, Spain

☺ These authors contributed equally to this work.
\* i.jimenez.gibd@gmail.com

**Data Availability Statement:** Data are available from DOI: 10.17632/ftnk8m3hdd.1 (http://dx.doi.org/10.17632/ftnk8m3hdd.1).

## Abstract

Foot orthoses are one of the most used strategies by healthy runners in injury prevention and performance improvement. However, their effect on running kinematics throughout an intense prolonged run in this population is unknown. Moreover, there is some controversy regarding the use of custom-made versus prefabricated foot orthoses. This study analysed the effect of different foot orthoses (custom-made, prefabricated and a control condition) on spatio-temporal and angular (knee flexion and foot eversion) kinematic parameters and their behaviour during an intense prolonged run. Twenty-four recreational runners performed three similar tests that consisted of running 20 min on a treadmill at 80% of their maximal aerobic speed, each one with a different foot orthosis condition. Contact and flight time, and stride length and stride rate were measured every 5 min by an optical measurement photoelectric cell system. Knee flexion and foot eversion kinematic parameters were measured by two high-speed cameras. No significant differences were found between the different foot orthoses in any of the time points studied and between the interaction of foot orthosis and behaviour over time, in any of the variables studied ($P > 0.05$). The use of custom-made and prefabricated foot orthoses during an intense prolonged run does not produce changes in spatio-temporal and kinematic parameters in healthy runners. These results suggest that a healthy runner maintains its ideal movement pattern throughout a 20 minute prolonged run, regardless the type of foot orthosis used.

## Introduction

Running is one of the most popular recreational and competitive physical activities due to its health benefits [1]. However, injury risk in the lower limb is also relatively high (20–79%) [2]. The use of foot orthoses is one of the most used strategies in the treatment of overuse running injuries [3,4]. Among the injured running population, the use of orthoses has been especially effective to runners with an excessive pronation or a length discrepancy [5], or pathologies

**Funding:** The work of IJP was supported by the Spanish Government (Ministerio de Ciencia, Innovación y Universidades, Grant FPU14//05626). In addition, the research project was supported by the Spanish Government (Ministerio de Economía y Competitividad, Subdirección General de Proyectos de Investigación. Convocatoria Proyectos I+D "Excelencia", Subprograma de Generación de Conocimiento, 2013 (project DEP2013-48420-P)). The funders had no role in study design, data collection and analysis, decision to publish, or preparation of the manuscript.

**Competing interests:** The authors have declared that no competing interests exist.

such as patellofemoral pain [6] or chronic Achilles tendon injury [7]. Despite the unclear evidence of the effect of foot orthoses in healthy recreational runners, its use is very usual as a mechanism of injury prevention and performance improvement [8–10]. In addition, runners perceive that footwear, where foot orthoses could be included, is one of the main factors associated with injury risk [11].

It has been observed that foot orthoses are able to reduce impact forces associated with running [12,13], to improve plantar pressure distribution [14,15], comfort perception [14,16] or to lead to kinematic changes [14,17,18]. However, the effect of foot orthoses during running has usually been studied through running trials, which may not be representative enough of the running load experienced by runners during their training sessions and competitions. In this sense, it would be interesting to investigate the effect of foot orthoses in a prolonged race, which should offer a more real vision of the recreational runners' usual practice [19]. Two previous studies of our research group [20,21] observed that foot orthoses have an effect on spatio-temporal, shock and plantar loading parameters after a prolonged run. The modification of those parameters could be associated with changes in kinematic parameters such as knee flexion and foot eversion [7,22], which are related with injury incidence [23], and justifies its investigation.

In this line, athletes may undergo kinematic modifications during a prolonged run in order to protect themselves from injuries, to maintain their performance level or as a result of fatigue [24]. In scientific literature, the most commonly kinematic adjustments observed are the increase in contact time, stride length, foot eversion and knee flexion [25–27]. By contrast, it has been speculated that foot orthoses could modify spatio-temporal kinematics [18] and angular kinematics during running due to the pronation motion control that they offer [7,8]. The type of foot orthosis could also have an influence in kinematic adjustments. Better results are commonly expected with custom-made foot orthoses than with prefabricated foot orthoses, due to its individual adaptation [28,29]. However, some authors have found a decrease in foot eversion with custom-made foot orthoses [9,14,30], while other studies have observed the same effect with prefabricated ones [17,31]. According to one of the last paradigms postulated by Nigg *et al.* [23,32], the use of footwear (or foot orthoses, in this case) should allow healthy runners to maintain their preferred movement path, so caution should be taken with kinematic alterations, since they will not always be positive. In this sense, there is a lack of information regarding the kinematic effects of foot orthoses during a prolonged run.

Therefore, the aim of the present study was to analyse the effect of different foot orthoses (custom-made, prefabricated and a control condition) on spatio-temporal and angular (knee flexion and foot eversion) kinematic parameters, as well as their behaviour during an intense prolonged run.

## Methods

### Participants

Twenty-four recreational runners: 18 males and 6 females (mean (standard deviation): age 34 (5) years; body mass 71.4 (12.5) kg; height 1.75 (0.08) m; running training distance 37.5 (12.8) kmweek) participated voluntarily in the present study. Inclusion criteria were: I) no history of lower extremity injuries within the last six months, II) no previous use of foot orthoses, and III) a training routine of at least 15 km/week. All runners signed the written informed consent before participation. The study procedures complied with the requirements established in the Declaration of Helsinki, and the study was approved by the University Ethics Committee (Comité Ético de Investigación en Humanos de la Comisión de Ética en Investigación Experimental de la Universitat de València, approval number H1427706182089).

### Foot orthosis conditions and customisation

Participants used three different foot orthosis conditions (Fig 1), on different days, and their order was previously randomised: 1) original insole of the shoe as a control condition; 2) pre-fabricated foot orthoses, chosen only according to athletes' foot size (Tecnoped Run, Herbitas, Valencia, Spain); and 3) custom-made foot orthoses built from a three-dimensional representation of the athlete's feet (OPCT Run, Sidas S.L., Barcelona, Spain). For the customisation of the custom-made foot orthoses, participants stood on a Printlab2 platform (Podiatech, Sidas Technologies, Voiron, France), composed of a pair of silicon vacuum bags that allowed the recreation of the plantar print. Based on the foot print and while the subastragaline joint was in a neutral position, a plaster mould was created, and through a thermo-welding process (Podiatech, Sidas Technologies, Voiron, France) the three dimensional foot orthoses customised to the participant's foot were built.

### Protocol

The study consisted of four tests on different days separated by a week: an initial field test and three laboratory tests, each one with a foot orthosis condition (Fig 2). The aim of the field test was to determine the individual maximal aerobic speed (MAS) in order to calculate the individual running speeds for the laboratory tests. In this first session, participants were asked to

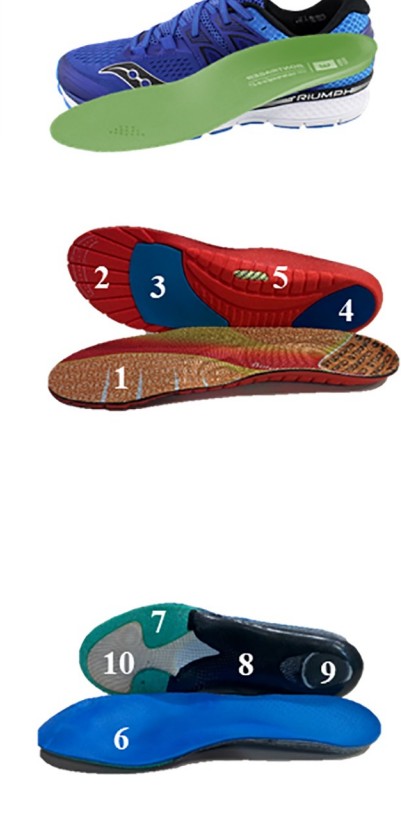

**Control**

Runner's training footwear

**Prefabricated**

- **Top layer (1):** anti-slip fabric Drytech.
- **Middle and lower layer (2):** antibacterial AirFoam.
- **Forefoot reinforcement (3):** polyurethane foam, hardness 15-25° - Propulsive airlatex.
- **Rearfoot reinforcement (4):** polyurethane foam, hardness 15-25° - Anti-shock airlatex.
- **Extra support under medial arch (5):** Technocarbon, 10 cm long and 3.5 cm high – Kevlar Carbon support.

**Custom-made**

- **Top layer (6):** Podiamic 160 micro-perforated polyethylene + ethyl-vinyl acetate (EVA), 2.5 mm thick, hardness 30° - hipoalergenic.
- **Forefoot insert (7):** Synthetic Viscotene® 2.5 mm thick, hardness 30° - propulsive properties.
- **Rearfoot insert (8):** podiaflex® resin 0.9 mm thick – cushioning and anti-shock properties.
- **Rearfoot reinforcement (9):** polyester resin Transflux®, 1 mm thick.
- **Sole reinforcement (10):** polyester resin Transflux®, 1 mm thick.

**Fig 1. Properties of the foot orthoses.**

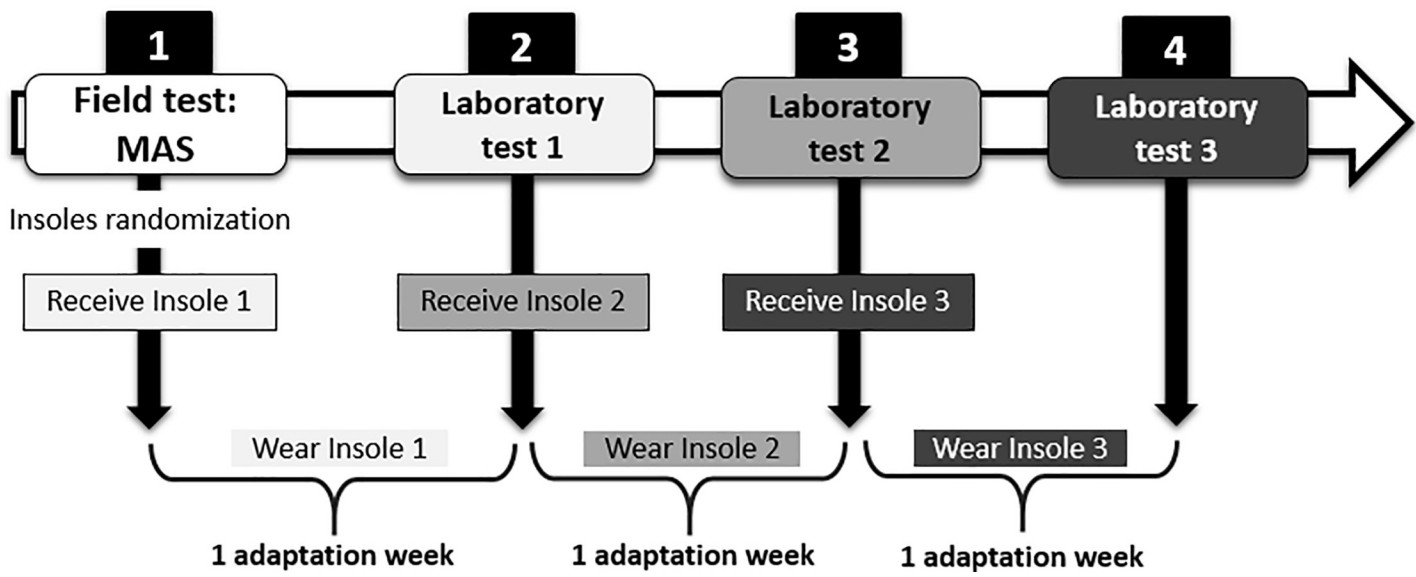

**Fig 2. Study design.**

run as maximum distance as possible during 5-min on a 400-m track and their individual MAS was determined from the total distance/time [33]. The three laboratory tests consisted of running 20 min on a treadmill (Excite Run 900, TechnoGymSpA, Gambettola, Italy) at 80% of their individual MAS, with a slope of 1%, to simulate the physiological load of an outdoor run [34]. Prior to these running tests, the athletes ran for 10 min at 60% of their MAS as a warm up exercise. Before each laboratory test, the runners trained for a week progressively with the foot orthosis assigned as an adaptation period [20]. In this adaptation period, participants were asked to wear the foot orthoses during the whole day for the first two days and, during the training sessions of these days, only in the warm-up and the cool down. From the third day onwards, if they did not experience discomfort, they had to wear the foot orthoses during their entire training session [35]. In addition, the participants wore their own running footwear in the three tests and in their adaptation periods in order to introduce no further changes in their running customary condition [36]. Perceived exertion was asked by means of the 20-point Borg scale [37], and heart rate was measured (Polar Electro Inc, Woodbury, USA), both at the beginning and the end of the test in order to control the test's intensity.

## Data collection and analysis

During each laboratory test, spatio-temporal kinematic parameters were collected with an optical measurement photoelectric cell system (OptoGait, Microgate, Bolzano, Italy) consisting of two bars located on the treadmill [38]. Contact time, flight time, stride length and stride rate were recorded for each foot every 5 min for 15 s during the main part of the running tests (20 min at 80% MAS). Data were extracted at 1000 Hz, and calculated with the OptoGait software program (Version 1.9.9.0, Microgate, Bolzano, Italy).

In addition, foot eversion and right knee flexion were analysed by two high-speed cameras (MotionScope®, Redlake, MASD Inc., San Diego, USA) sampling at 125 Hz. One camera was placed 1.5 m perpendicular to the motion plane and 0.5 m high to measure the frontal plane, and the other camera was placed 1.5 m perpendicular to the right motion plane and 1 m high to measure the sagittal plane. Angular kinematic data were recorded simultaneously with the

Optogait, every 5 min for 5 s during the 20 min run. Both cameras were synchronised with the camera software (Redlake MASD MotionScope®, San Diego, USA), and data were analysed using a motion analysis software (Kinescan/IBV System, Valencia, Spain) [39]. Prior to each measurement, optical distortion of the camera lens and calibration of the space were performed using a square object of known dimensions in which four space references were attached. Calibration was performed via bidimensional (2D) direct linear transformation using the motion analysis software. The spline smoothing method was used automatically in the motion analysis software [40].

For the kinematic analysis of the right knee flexion a 2D model of 3 reflective spherical markers (diameter: 16 mm) was used as in previous studies [41]. Markers were placed in the right leg: 1) on the greater trochanter of the femur, 2) on the lateral femoral epicondyle of the knee, and 3) on the lateral malleolus of the ankle. Knee flexion was calculated by the projected α angle between the two segments (thigh and leg) defined by the kinematic model [41] (Fig 3). In addition, foot eversion of both feet was analysed using Clarke *et al.*'s 2D model of four markers [42,43] placed: 1) on the gastrocnemius (in the axial line of the leg, under the gastrocnemius bifurcation), 2) on the Achilles tendon (at the height of the malleolus), 3) on the upper part of the calcaneus, and 4) on the lower part of the calcaneus. The projected β angle between the two segments (calcaneus and leg) defined by the kinematic model [12,43] was used to calculate foot eversion (Fig 3). To normalise the values, knee flexion and foot eversion angles were calculated from a static standing trial wearing the assigned foot orthoses condition, considered as 0° [41,43].

The variables of interest were: knee flexion at contact time, maximum knee flexion during stance phase, knee flexion at toe-off, maximum knee flexion during swing phase, foot eversion at contact time, and maximum foot eversion during the stance phase. These variables were calculated from the average of 5 steps [7,41]. Angles during toe-off and swing phase were measured due to the anti-shock and propulsive properties suggested by the foot orthoses manufacturer.

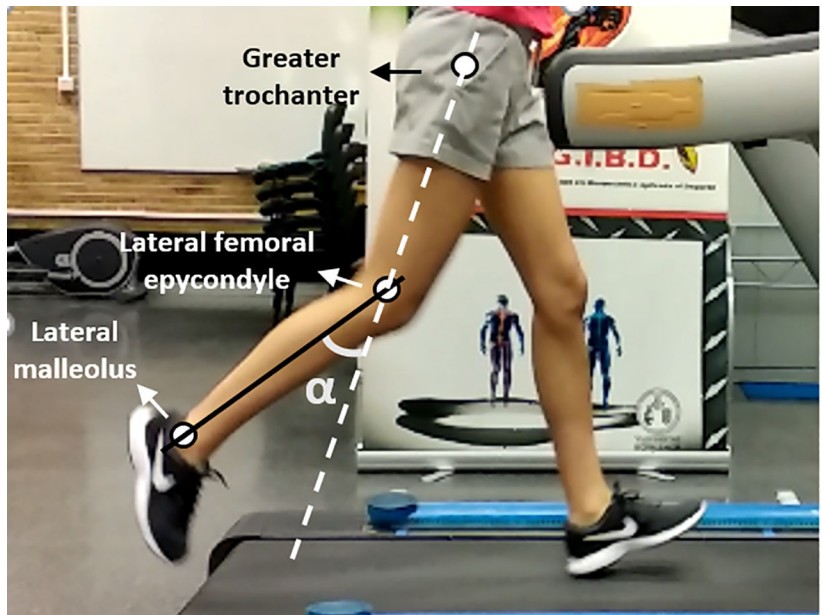
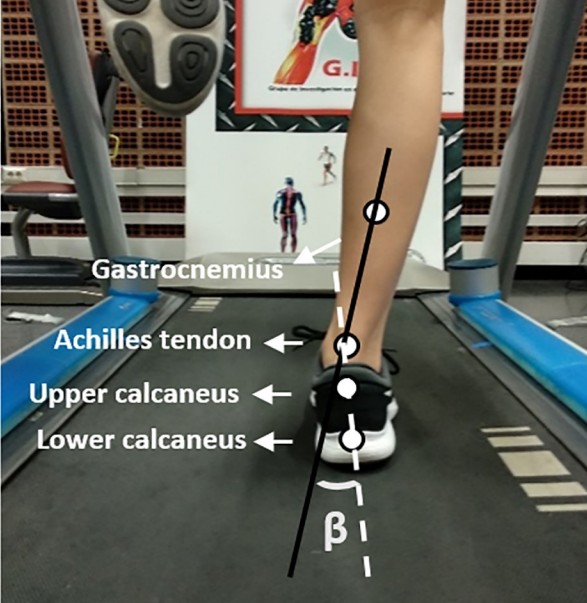

**Fig 3.** Placement of markers and kinematic 2D models used to measure α angle of knee flexion (left) and β angle of foot eversion (right) during running.

## Statistical analysis

For the statistical analysis, data were analysed using a statistical package (SPSS 20.0, IBM, Armonk, USA). After confirming the normality of the sample in all the variables by the Shapiro Wilks test ($P > 0.05$), and verifying the sphericity by the Mauchly Sphericity test, the descriptive statistics were extracted. Data were reported as mean (standard deviation (SD)). Then, 2-way repeated-measures ANOVAs were performed for each dependent variables of the study. Two factors were used as intra-subject factors: foot orthosis condition, with three levels (control, prefabricated and custom-made) and time points with 2 levels in rate of perceived exertion and heart rate (minute 1 and minute 19) and 5 levels in kinematic parameters (minutes 1, 5, 10, 15 and 20). For the significant ANOVA models ($P < 0.05$), the Bonferroni correction post-hoc test was carried out. For the pair significant differences ($P < 0.05$), the Cohen's effect size (*ES*) was computed and classified as small (*ES* 0.2–0.5), moderate (*ES* 0.5–0.8) or large (*ES* > 0.8) [44]. Significance level was stablished at $\alpha = 0.05$.

## Results

Rate of perceived exertion and heart rate values are presented in Table 1. For rate of perceived exertion and heart rate, no significant main effect of condition ($P > 0.22$) or time-by-condition interaction effect ($P > 0.21$) was found. However, there was a main effect of time ($P < 0.001$; $ES > 1.0$). These results indicate an increase of rate perceived exertion and heart rate at the end of the run regardless of the foot orthosis condition.

No differences were observed between feet in spatio-temporal and foot eversion parameters ($P > 0.05$). Therefore, foot dominance factor was not considered in the repeated measures ANOVA and the mean of both feet is presented for these variables.

Tables 2–4 show the behaviour of spatio-temporal, knee flexion and foot eversion parameters, respectively, during the running test on each foot orthosis condition. There was no significant main effect of condition ($P > 0.36$) and no significant effect in the interaction time-by-condition ($P > 0.18$) in any of the kinematic variables (spatio-temporal, knee flexion and foot eversion parameters) except for a main effect of time only in contact time and maximum eversion during the stance phase ($P < 0.01$). In contact time, this result implies higher values in minute 10 vs minute 1 ($P = 0.02$; $ES = 0.2$) and in minute 20 vs minute 1 ($P = 0.02$; $ES = 0.2$), both of them with a small effect size. Moreover, in maximum eversion during the stance phase, this result implies higher values at each time point studied with respect to minute 1 (min 1 vs min 5 ($P < 0.001$; $ES = 0.2$); vs min 10 ($P < 0.001$; $ES = 0.2$); vs min 15 ($P = 0.001$ $ES = 0.2$); vs min 20 ($P < 0.001$ $ES = 0.3$), showing in all cases a small effect size. In all the other variables, no significant main effect of time was found ($P > 0.44$).

Percent changes of the biomechanical variables for each participant as well as the number of participants that showed a biomechanically relevant change (±10%) [36] regarding the control condition are provided in the Supplementary File.

**Table 1. Perceived exertion rate and heart rate differences between time points (Begin vs. End) in each foot orthosis condition.**

|  | Control | | Prefabricated | | Custom-made | |
|---|---|---|---|---|---|---|
|  | Mean (SD) | | Mean (SD) | | Mean (SD) | |
|  | Begin | End | Begin | End | Begin | End |
| Perceived exertion rate (points) | 12.7 (1.2) | 14.3 (1.7) | 12.8 (1.1) | 14.7 (1.7) | 13.0 (1.5) | 15.0 (2.0) |
| Heart rate (bpm) | 162.8 (9.3) | 171.9 (9.1) | 158.6 (10.5) | 170.3 (10.8) | 160.1 (11.2) | 172.3 (12.0) |

**Table 2. Spatio-temporal parameters during the 5 time points of the running test with the 3 foot orthosis conditions.**

| Variable | Minute | Control | Prefabricated | Custom-made |
|---|---|---|---|---|
| | | *Mean (SD)* | *Mean (SD)* | *Mean (SD)* |
| Stride Length (m) | 1 | 2.45 (0.31) | 2.48 (0.33) | 2.47 (0.32) |
| | 5 | 2.45 (0.32) | 2.48 (0.33) | 2.48 (0.34) |
| | 10 | 2.46 (0.32) | 2.48 (0.32) | 2.47 (0.33) |
| | 15 | 2.46 (0.31) | 2.48 (0.32) | 2.48 (0.33) |
| | 20 | 2.47 (0.32) | 2.47 (0.30) | 2.44 (0.40) |
| Stride Rate (Hz) | 1 | 1.43 (0.06) | 1.42 (0.07) | 1.43 (0.07) |
| | 5 | 1.43 (0.06) | 1.42 (0.07) | 1.42 (0.07) |
| | 10 | 1.42 (0.07) | 1.42 (0.07) | 1.42 (0.07) |
| | 15 | 1.42 (0.06) | 1.42 (0.07) | 1.41 (0.07) |
| | 20 | 1.41 (0.06) | 1.43 (0.07) | 1.41 (0.07) |
| Contact Time (s) | 1 | 0.268 (0.023) | 0.268 (0.024) | 0.269 (0.024) |
| | 5 | 0.271 (0.023) | 0.271 (0.024) | 0.271 (0.025) |
| | 10 | 0.272 (0.021) | 0.273 (0.025) | 0.273 (0.021) |
| | 15 | 0.271 (0.021) | 0.275 (0.024) | 0.273 (0.022) |
| | 20 | 0.273 (0.022) | 0.273 (0.024) | 0.275 (0.025) |
| Flight Time (s) | 1 | 0.083 (0.026) | 0.085 (0.029) | 0.082 (0.026) |
| | 5 | 0.081 (0.026) | 0.081 (0.028) | 0.081 (0.030) |
| | 10 | 0.080 (0.025) | 0.080 (0.029) | 0.080 (0.026) |
| | 15 | 0.083 (0.023) | 0.078 (0.028) | 0.081 (0.026) |
| | 20 | 0.080 (0.027) | 0.079 (0.026) | 0.080 (0.030) |

No differences were observed in the interaction between foot orthosis condition and time point ($P > 0.05$).

## Discussion

The purpose of this study was to analyse the effects of custom-made and prefabricated foot orthoses on spatio-temporal and angular kinematic parameters during an intense prolonged run. The main result of the study was that there was not a significant main effect of foot orthosis or an interaction between foot orthosis and time on the spatio-temporal and angular kinematic variables assessed during running.

It is well known that during a high intensity run, and consequently, with the development of it, some kinematic alterations previously associated with injury and performance optimisation can occur [24,27]. In this line of thought, it has been suggested that the use of foot orthoses may be a great strategy to counteract those changes, for example through greater control of foot pronation [17,45]. In this sense, in the present work the effect on spatio-temporal and angular kinematic parameters of different foot orthoses were analysed in different time points of an intense prolonged run of 20 minutes of duration, in order to know the behaviour of those parameters throughout this time in a more ecological protocol [19]. Currently, there is little information on the effectiveness of foot orthoses during an intense run. Only the studies of Lucas-Cuevas *et al.* [20,21] analysed some spatio-temporal parameters in a continuous running protocol, with three different foot orthosis conditions (similar to those used in the present work), but only at the beginning and at the end of the run, without looking at their behaviour throughout the run.

In the present study, and according to the results of these two investigations [20,21], no significant differences in several spatio-temporal parameters under three different conditions of foot orthoses as well as in the interaction between foot orthosis and running time were

**Table 3. Knee flexion parameters during the 5 time points of the running test with the 3 foot orthosis conditions.**

| Variable | Minute | Control | Prefabricated | Custom-made |
|---|---|---|---|---|
| | | *Mean (SD)* | *Mean (SD)* | *Mean (SD)* |
| Knee flexion at contact time (˚) | 1 | 12.17 (4.46) | 11.45 (5.69) | 12.19 (4.44) |
| | 5 | 12.11 (4.67) | 11.89 (6.01) | 11.89 (4.08) |
| | 10 | 12.30 (5.14) | 11.73 (6.37) | 12.28 (4.23) |
| | 15 | 12.12 (4.63) | 11.53 (5.76) | 12.73 (5.05) |
| | 20 | 11.86 (4.74) | 11.58 (5.99) | 12.23 (5.19) |
| Maximum knee flexion during stance phase (˚) | 1 | 39.31 (4.70) | 38.53 (5.77) | 39.01 (4.37) |
| | 5 | 39.12 (4.91) | 38.73 (5.69) | 38.99 (4.13) |
| | 10 | 38.65 (4.56) | 38.98 (5.40) | 39.45 (4.45) |
| | 15 | 38.69 (4.51) | 38.72 (6.32) | 39.70 (4.72) |
| | 20 | 38.02 (5.04) | 38.90 (5.40) | 39.05 (5.13) |
| Knee flexion at toe-off (˚) | 1 | 10.78 (4.77) | 11.19 (5.61) | 11.03 (5.65) |
| | 5 | 11.01 (4.90) | 10.97 (5.84) | 10.27 (5.52) |
| | 10 | 9.82 (5.05) | 10.74 (5.77) | 10.69 (5.69) |
| | 15 | 10.22 (5.18) | 10.95 (6.05) | 10.97 (6.23) |
| | 20 | 10.04 (5.00) | 11.35 (6.14) | 10.70 (5.90) |
| Maximum knee flexion during swing phase (˚) | 1 | 87.74 (10.52) | 88.49 (10.94) | 86.92 (10.66) |
| | 5 | 88.59 (10.24) | 87.40 (9.92) | 88.18 (10.95) |
| | 10 | 87.34 (10.39) | 88.23 (9.71) | 87.54 (10.67) |
| | 15 | 88.26 (10.63) | 88.29 (9.94) | 87.63 (10.31) |
| | 20 | 88.46 (11.18) | 87.73 (9.50) | 87.35 (11.23) |

No differences were observed in the interaction between foot orthosis condition and time point ($P > 0.05$).

observed. A previous study has shown that runners are able to achieve a lower metabolic expenditure by adjusting these parameters in an unconscious and individual way [46]. Therefore, these results could explain that the intervention with foot orthoses does not involve a greater energy expenditure, because it is not necessary to modify these parameters with respect to the control condition during an intense prolonged run of 20 minutes. In addition, running with foot orthoses could be affecting other variables such as ground reaction forces,

**Table 4. Foot eversion parameters during the 5 time points of the running test with the 3 foot orthosis conditions.**

| Variable | Minute | Control | Prefabricated | Custom-made |
|---|---|---|---|---|
| | | *Mean (SD)* | *Mean (SD)* | *Mean (SD)* |
| Foot eversion at contact time (˚) | 1 | -5.97 (5.79) | -6.50 (6.41) | -6.10 (7.00) |
| | 5 | -5.92 (6.26) | -6.53 (6.85) | -6.25 (7.39) |
| | 10 | -5.71 (6.14) | -6.27 (7.10) | -6.36 (7.13) |
| | 15 | -6.22 (6.45) | -6.42 (6.99) | -6.73 (7.68) |
| | 20 | -6.01 (6.09) | -6.18 (7.15) | -6.61 (6.98) |
| Maximum foot eversion during stance phase (˚) | 1 | 10.98 (4.16) | 10.30 (3.85) | 10.66 (4.23) |
| | 5 | 11.66 (4.00) | 11.21 (4.18) | 11.43 (4.38) |
| | 10 | 11.56 (3.76) | 11.53 (4.99) | 11.64 (4.70) |
| | 15 | 11.91 (3.88) | 11.30 (4.13) | 11.59 (4.65) |
| | 20 | 11.87 (4.09) | 11.76 (4.36) | 11.70 (4.92) |

No differences were observed in the interaction between foot orthosis condition and time point ($P > 0.05$).

neuromuscular activation or comfort. Previous studies have observed that the use of foot orthoses seems to reduce impacts and to improve the comfort perception of runners in relation to not wearing them [20,21,30]. In this line of thought, foot orthoses could be prescribed seeking these benefits without compromising the athlete's running technique during an intense prolonged run. Therefore, future studies should address the effect of foot orthoses from a holistic perspective that include the combined assessment of kinematics, comfort, neuromuscular activation, and ground reaction forces.

In the present study, none of the variables of knee flexion was altered throughout the run when using foot orthoses (custom-made and prefabricated) in comparison with the control condition. These results are in agreement with previous studies performed in trials [9,13,45]. This lack of modifications found in knee flexion could be attributed to the idea that the running movement is pre-programmed, and in healthy runners the running pattern normally used could already be the ideal one, without needing an alteration of the same [23,32]. So, its change would lead to an increase in metabolic expenditure [23,32]. Therefore, as the athletes in this study were injury-free, they could have adapted the use of the foot orthoses to their pre-established movement pattern during the training period with them.

In addition, according to Nigg *et al.* [23,32], an intervention with foot orthoses would be beneficial if it allows to maintain the ideal and preferred movement path. In this line, Lucas-Cuevas *et al.* [20] used the same foot orthoses and a similar sample population as in the present investigation, and these authors observed lower acceleration rates with the custom-made orthoses compared to the prefabricated ones. This reduction in the acceleration rate, along with the lack of modifications in the knee flexion observed by the present study, could suggest a higher level of protection by the custom-made foot orthoses, as their use would avoid unnecessary alterations in the movement pattern and the kinematic chain to reduce the impacts, which could be harmful [23].

With respect to foot eversion, a large number of studies have analysed the effect of foot orthoses aiming to control pronation, but only in trials or brief run protocols and ultimately finding different results. In the present study, the use of foot orthoses did not produce any change in the maximum foot eversion during stance phase or in the foot eversion at contact time in any of the analysed time points, in comparison with the control condition. Similarly, previous studies with healthy runners also found no changes in foot eversion [36,47]. On the contrary, in studies looking into a running population with pathologies, such as patellofemoral syndrome, anterior knee pain and chronic Achilles tendon injury, changes in foot eversion were observed as a result of using foot orthoses [7,45,48]. These findings suggest that the type of population may influence the variability of results. Therefore, the lack of modifications in these parameters could be explained, as mentioned in the knee flexion, by the fact of having studied athletes without injuries, who would not need to use a rearfoot motion control strategy and who were able to maintain their ideal running pattern during the 20 minutes of run. In addition, while previous authors found decreases in foot eversion with both custom-made [9,14,30], and prefabricated foot orthoses [17,31], no differences were found in this study between the two types of foot orthoses. Therefore, other hypotheses that could justify these results are the specific characteristics of the foot orthoses, which were not specifically manufactured to limit the movement of the ankle [12]; and the protocol used to measure these parameters [8,49].

Regarding the running protocol used in this study, participants reported perceived exertion values between 13 and 15 points as well as heart rate values between 158 and 172 bpm, at the beginning and at the end of the tests, respectively. These values correspond to a somewhat hard/hard intensity, and show that athletes performed the 20 min tests at the anaerobic threshold level [50]. In a previous study [51], no differences were observed between the three foot

orthosis conditions in any of the two parameters, neither at the beginning nor at the end of the running protocol. In the present study, however, there was a significant increase in perceived exertion and heart rate at the end of the running protocol in the three foot orthosis conditions, which shows that there was a progression in the intensity of the effort throughout the running protocol.

Despite this fact, in this study no kinematic changes were observed at the different time points of the running test. There was only a main effect of time in contact time and maximum eversion during the stance phase, regardless of foot orthosis condition, which implied certain increases in these variables as the athletes were completing the running tests, with small effect sizes. This lack of changes throughout time could be due to an insufficient level of fatigue. In this sense, the athletes of this study, despite having worked at the anaerobic threshold, may have not reach a sufficient level of muscular fatigue to produce changes in the movement pattern [20]. As a result, it is possible that the modifications or compensatory adaptations could have taken place at the neuromuscular level, without showing changes in the kinematic pattern of movement. Therefore, the intensity of the running tests, which was not hard or long enough to produce movement pattern alterations throughout the run, could be suggested as a possible limitation of the study. Future studies should assess the effect of custom-made and prefabricated foot orthoses on kinematic parameters after longer or more intense protocols.

The main limitation of this study was the use of 2D video to analyse running kinematics instead of using the "gold standard" 3D video analysis. However, it has been stated that 2D video analysis is accurate enough to assess frontal plane variables of treadmill running [52]. Likewise, the study of a single plane in both the knee and the ankle could also be considered a limitation, since other planes could have shown kinematic modifications, as in the knee abduction and adduction movements [49,53]. Another limitation to take into account was the small number of women that took part in the study. A larger sample of female runners would have enabled the analysis of movement patterns by gender, as it is known that women have structural differences which can lead to differences in running mechanics [54] and to a different influence of foot orthoses.

From this point of view, although no differences have been found between the three foot orthosis conditions studied from a grouped perspective, in the studies of foot orthosis and footwear it is common to find a large variability of response among participants, which is masked by the overall results [36]. In the present study, as presented in the Supplementary File, the spatio-temporal parameters have not been affected by inter-subject variability, since most participants have not experienced a relevant change. However, in the angular parameters, the effects of different directions and magnitudes have been observed both among participants as well as within a single individual. In the future, it would be interesting to study if these different responses to a foot orthosis implementation could be grouped by some participant characteristics, such as the gender or the type of foot.

In addition, in future studies it would be interesting to investigate the effect of this foot orthosis intervention in a pathological population or with pain, in order to know if the use of foot orthosis may have some effect in such pathology, with a consequent kinematic alteration. Furthermore, it would also be interesting to analyse the patterns of movement by gender, with an equivalent sample, by the aforementioned differences between men and women. Finally, the study of the long-term effects of a similar intervention of foot orthoses could be of interest in order to understand the influence over time of using orthoses on running mechanics, even with the creation of two groups: one control and another using specific foot orthoses.

## Conclusions

The findings of this study show that the use of custom-made and prefabricated foot orthoses during an intense prolonged run does not produce changes in spatio-temporal and angular (knee flexion and foot eversion) kinematic parameters, compared to a control condition. This lack of modifications in the running mechanics of the athletes may suggest that a healthy runner population is able to maintain their ideal movement pattern throughout a 20 minute prolonged run, regardless of the type of foot orthosis used.

## Supporting information

**S1 Fig. Percent changes relative to control foot orthosis condition for spatio-temporal parameters regardless the time points of the running test.**
(JPG)

**S2 Fig. Percent changes relative to control foot orthosis condition for knee flexion parameters regardless the time points of the running test.**
(JPG)

**S3 Fig. Percent changes relative to control foot orthosis condition for foot eversion parameters regardless the time points of the running test.**
(JPG)

**S1 Table. Number of participants experiencing biomechanically relevant reductions (≥10% Reduction), biomechanically relevant increases (≥10% increase), and no change (change between -9.9% and +9.9%) for each variable, regardless the time points of the running test, when wearing prefabricated or custom-made foot orthoses compared to control condition.**
(DOCX)

## Acknowledgments

We thank all the participants for their voluntary participation in this study.

## Author Contributions

**Conceptualization:** Marina Gil-Calvo, Jose Ignacio Priego-Quesada, Ángel G. Lucas-Cuevas, Pedro Pérez-Soriano.

**Data curation:** Marina Gil-Calvo, Irene Jimenez-Perez.

**Formal analysis:** Marina Gil-Calvo, Irene Jimenez-Perez, Jose Ignacio Priego-Quesada.

**Funding acquisition:** Marina Gil-Calvo, Irene Jimenez-Perez.

**Investigation:** Marina Gil-Calvo, Irene Jimenez-Perez.

**Methodology:** Marina Gil-Calvo, Irene Jimenez-Perez, Jose Ignacio Priego-Quesada, Ángel G. Lucas-Cuevas, Pedro Pérez-Soriano.

**Project administration:** Marina Gil-Calvo, Pedro Pérez-Soriano.

**Resources:** Marina Gil-Calvo, Pedro Pérez-Soriano.

**Software:** Irene Jimenez-Perez, Jose Ignacio Priego-Quesada.

**Supervision:** Marina Gil-Calvo, Jose Ignacio Priego-Quesada, Ángel G. Lucas-Cuevas, Pedro Pérez-Soriano.

**Validation:** Marina Gil-Calvo, Irene Jimenez-Perez, Jose Ignacio Priego-Quesada, Ángel G. Lucas-Cuevas, Pedro Pérez-Soriano.

**Visualization:** Marina Gil-Calvo, Irene Jimenez-Perez, Jose Ignacio Priego-Quesada, Ángel G. Lucas-Cuevas, Pedro Pérez-Soriano.

**Writing – original draft:** Marina Gil-Calvo, Irene Jimenez-Perez, Jose Ignacio Priego-Quesada, Ángel G. Lucas-Cuevas, Pedro Pérez-Soriano.

**Writing – review & editing:** Marina Gil-Calvo, Irene Jimenez-Perez, Jose Ignacio Priego-Quesada, Ángel G. Lucas-Cuevas, Pedro Pérez-Soriano.

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
