## [Decision Letter · Decision Letter 0]

9 Dec 2019

PONE-D-19-28910

Effect of custom-made and prefabricated foot orthoses on kinematic parameters during an intense continuous run

PLOS ONE

Dear Ms. Jimenez-Perez,

Thank you for submitting your manuscript to PLOS ONE. After careful consideration, we feel that it has merit but does not fully meet PLOS ONE’s publication criteria as it currently stands. Therefore, we invite you to submit a revised version of the manuscript that addresses the points raised during the review process.

In the revised manuscript, you need to address all of the comments posed by the reviewers. Specifically, please provide more focused introduction and discussion sections, discuss individual variation in relation to the present study, relate the paper to the effects on a prolonged run, and justify the subject population.

We would appreciate receiving your revised manuscript within 60 days. To enhance the reproducibility of your results, we recommend that if applicable you deposit your laboratory protocols in protocols.io, where a protocol can be assigned its own identifier (DOI) such that it can be cited independently in the future. For instructions see: http://journals.plos.org/plosone/s/submission-guidelines#loc-laboratory-protocols

We look forward to receiving your revised manuscript.

Kind regards,

Alena Grabowski

Academic Editor

PLOS ONE

Journal Requirements:

2. Thank you for including your ethics statement; "The study procedures complied with the requirements established in the Declaration of Helsinki, and the study was approved by the University Ethics Committee (approval number H1427706182089). All participants signed a written informed consent before participation."

3.  We note that your study is closely related to the following publication, on which you are an author:

https://doi.org/10.1371/journal.pone.0173179

Although you have cited the above study in the discussion section of your article, we feel that the scientific rationale of the current study and the contribution that it makes to the field should be better justified.  Therefore, please cite and discuss the above study in the introduction section of your manuscript, clarifying how the present work is related to the previously published paper.

Please note that our second publication criterion states that "If a submitted study replicates or is very similar to previous work, authors must provide a sound scientific rationale for the submitted work and clearly reference and discuss the existing literature. Submissions that replicate or are derivative of existing work will likely be rejected if authors do not provide adequate justification." http://www.plosone.org/static/publication.action#results.

Reviewers' comments:

Reviewer's Responses to Questions

**Comments to the Author**

1. Is the manuscript technically sound, and do the data support the conclusions?

Reviewer #1: Yes

Reviewer #2: Yes

2. Has the statistical analysis been performed appropriately and rigorously? 

Reviewer #1: Yes

Reviewer #2: Yes

3. Have the authors made all data underlying the findings in their manuscript fully available?

Reviewer #1: Yes

Reviewer #2: Yes

4. Is the manuscript presented in an intelligible fashion and written in standard English?

Reviewer #1: Yes

Reviewer #2: Yes

5. Review Comments to the Author

Reviewer #1: Overall/General comments and primary concerns:

Given that the purpose of the study is to investigate the effects of orthoses during a prolonged run, I recommend that the authors focus the introduction to describe what kinematic changes do occur over the course of a prolonged run and what benefits orthoses may provide. The current focus of the introduction is to describe the more general research surrounding orthoses so the inclusion of the prolonged running seems to me more of an afterthought than a focus.

The introduction should also describe and explain why healthy runners are an appropriate population to study for this investigation given that orthoses were originally designed to help patient populations. I suggest that the authors should discuss runner’s use of these products to prevent injury. The survey study in JOSPT on runner’s beliefs about injury may be helpful in this regard. Please also provide a rationale for examining knee flexion and be more explicit about why foot eversion was studied (I imagine that the reason is that orthoses are intended to modify eversion primarily, but whatever the reason should be stated).

It did not appear that the authors examined the individual subject data. Footwear and foot orthoses research is notorious for finding insignificant differences, not because the conditions have no effect but because the inter-individual variation in the response to the conditions is so large.

The discussion was a bit outside of the scope of the present study in some areas (for example, with respect to energy expenditure and what affect that may have on the results. The present study only examined HR and RPE, neither of which were between orthosis conditions). Any discussion regarding energy expenditure or running economy should be brief, to the point, and within the scope of the present study (i.e. one paragraph of no more than 10-12 lines in length). The scope of the paragraph discussing the knee flexion results (line 299-319) is a good example of the scope that should be used to explain the other variables and length should not exceed that of this example section, given the results and the overall scope of the study.

Similar to the introduction, the discussion section seems to focus on the differences between conditions in the primary outcome variables (knee angle, eversion angle, and spatio-temporal parameters), rather than the effects of the orthoses on the prolonged run. I understand that it is difficult to explain the effects that the orthoses had on the prolonged run given that there were no significant differences observed. However, the purpose of this suggestion is make sure the discussion surrounding the prolonged run aspect of this study is not seen as an afterthought; it should be the focus of both the intro and discussion because it is the primary difference compared with previous studies.

Abstract

1. Line 21-23: I recommend revising this section of the abstract to reflect better why healthy runners would chose to wear foot orthoses or shoe inserts and thus why they are a good sample to study rather than a patient population.

2. Line 36-38: I recommend changing the last sentence of the conclusion to be something that can be said about the present study. Additional work can always be done but a more impactful conclusion would reflect what can be learned by the present study. As discussed above, commenting about inter-individual differences may be a good topic to replace this last sentence (see further comment below about the inter-individual differences).

Introduction

3. Line 45: Please be specific here as to not unintentionally spread speculation. The referenced study seems to be about Achilles tendon injuries alone. Also, please confirm that the findings of this study were indeed that the orthoses prevented injury, which is the statement in the text of the present study. Citations should reference the findings, not the speculation or theories proposed by others.

4. Line 60: “…no clear evidence…of the effect of foot orthoses on running kinematics” is not clear because the authors described some kinematic variables earlier in this paragraph. Please be more specific as to what is still unknown. Also in this paragraph, the ‘preferred movement path’ is discussed as a follow-up to describing that there is not a lot of evidence regarding spatio-temporal variables. Although both portions of the sentence are true individually, they should not be together because the preferred movement path is less a consideration for spatio-temporal characteristics, and more about joint and segment kinematics.

Methods

5. The authors described that they tested if the variables were normally distributed but did not describe the results of these tests in the statistical analysis section of the methods or in the results. Please describe which variables, if any, were non-normally distributed and what was done (if anything) to adjust for non-normal data.

Results

6. Line 215-216: The end line 216 which is part of the sentence, “there was no significant main effect…in any of the kinematic variables; however, there was a main effect for…” should be changed to “…in any of the kinematic variables except for a main effect for…”. More simply stated, please change “however” to “except for”

7. As described in the general/overall comments, it may be beneficial to examine the inter-individual differences.

Discussion

8. Line 250: The findings of the referenced study were not “harmful kinematic alterations.” We do not have enough longitudinal evidence to imply that any cross-sectional study is harmful.

9. Line 258: These are the only studies that looked at spatio-temportal parameters with respect to what? Orthoses? These are far from the only prolonged running studies out there. This sentence should be more specific.

10. Line 275: Need to explain how the methods of referenced study (Wilkinson et al) are similar and different than the present study so the reader can have more context to understand why the results between the two studies are being compared and understand the potential explanation for the differences. Start the paragraph explaining this information rather than bring it in later (i.e. line 279-283).

Figures and tables

11. There is a typo with the spelling of ‘length’ in Figure 4

12. It is my personal preference to have the full variable name used as figure titles instead of the abbreviation. That way, I don’t have to remember or look back at what something is, I know it the instant that I look at it.

Reviewer #2: Overall, the manuscript is well-written, with a few areas where the use of English grammar is not quite up to expectations. The authors do well to provide background review of the literature and justification for the study. It seems the gap that the authors hope to fill is the effect of custom orthoses over time-course of an intense continuous run. However, the findings of the study indicate that there was no statistical difference to report. A null finding is not considered a negative thing, but it does substantially reduce the enthusiasm for this study. Additionally, while the authors have provided a thorough treatment of the data and kept their conclusions within the scope of the statistical results, they do tend to overstate potential application of this work in the Conclusion statement. I'm not sure that enough evidence was presented to say that impact reduction with orthoses can be a benefit "without interfering in the running technique or running economy".

There are a large number of figures, which are unfortunately not very informative. Figures 1-3 are generally helpful to explain the methods, but the information in Figures 4-6 could easily be placed in a table, or in supplementary material, as they really do not show anything more than what the authors have stated in the text.

The discussion section is excessively long, with too much revisitation of the previous work. Rather than re-reviewing all of the literature here, the manuscript would be improved with a critical evaluation of the points in the literature which may help explain why the current study resulted in null findings.

6. PLOS authors have the option to publish the peer review history of their article (what does this mean?). If published, this will include your full peer review and any attached files.

Reviewer #1: No

Reviewer #2: No

---

## [Author Response · Author response to Decision Letter 0]

14 Jan 2020

Thank you for submitting your manuscript to PLOS ONE. After careful consideration, we feel that it has merit but does not fully meet PLOS ONE’s publication criteria as it currently stands. Therefore, we invite you to submit a revised version of the manuscript that addresses the points raised during the review process.

In the revised manuscript, you need to address all of the comments posed by the reviewers. Specifically, please provide more focused introduction and discussion sections, discuss individual variation in relation to the present study, relate the paper to the effects on a prolonged run, and justify the subject population.

Dear editor, we are grateful for the opportunity to revise our paper. All the comments sent by the reviewers were carefully addressed. All the changes in the edited manuscript are highlighted with red text in the revised submission. We look forward to hearing your editorial decision.

Journal Requirements:

Reply: During the review, the style requirements of PLOS ONE’s have been followed.

2. Thank you for including your ethics statement; "The study procedures complied with the requirements established in the Declaration of Helsinki, and the study was approved by the University Ethics Committee (approval number H1427706182089). All participants signed a written informed consent before participation."

Reply: We would like to thank the editor for this comment. We have include the full name of the ethics committee in Methods section of the manuscript, and to the “Ethics Statement” field of the submission form, as follow (Page 5, Lines 96-100): 

“The study procedures complied with the requirements established in the Declaration of Helsinki, and the study was approved by the University Ethics Committee (Comité Ético de Investigación en Humanos de la Comisión de Ética en Investigación Experimental de la Universitat de València, approval number H1427706182089).”

3. We note that your study is closely related to the following publication, on which you are an author:

https://doi.org/10.1371/journal.pone.0173179

Although you have cited the above study in the discussion section of your article, we feel that the scientific rationale of the current study and the contribution that it makes to the field should be better justified. Therefore, please cite and discuss the above study in the introduction section of your manuscript, clarifying how the present work is related to the previously published paper.

Please note that our second publication criterion states that "If a submitted study replicates or is very similar to previous work, authors must provide a sound scientific rationale for the submitted work and clearly reference and discuss the existing literature. Submissions that replicate or are derivative of existing work will likely be rejected if authors do not provide adequate justification." http://www.plosone.org/static/publication.action#results.

Reply: We would like to thank the editor for this comment. We have cited and discussed this publication in the introduction section, to clarify that it is a previous study of our research group, as suggested (Page 3, Lines 60-65): 

“Two previous studies of our research group [20,21] observed that foot orthoses have an effect on spatio-temporal, shock and plantar loading parameters after a prolonged run. The modification of those parameters could be associated with changes in kinematic parameters such as knee flexion and foot eversion [7,22], that are related with the injury incidence [23], which justifies its investigation.”

Reply: Yes, it is right, after acceptance, we will provide the DOI of the repository information of our data. 

Reviewers' comments:

Reviewer #1: Overall/General comments and primary concerns:

Given that the purpose of the study is to investigate the effects of orthoses during a prolonged run, I recommend that the authors focus the introduction to describe what kinematic changes do occur over the course of a prolonged run and what benefits orthoses may provide. The current focus of the introduction is to describe the more general research surrounding orthoses so the inclusion of the prolonged running seems to me more of an afterthought than a focus.

Reply: We would like to thank the reviewer for this recommendation. We have rearranged the introduction to focus it from the beginning on the effect of the prolonged run. Moreover, we have put together the description of the kinematic changes of a prolonged run and the benefits that foot orthoses can provide, to give a clearer vision of the study purpose.

In addition, throughout the manuscript the term “continuous run” has been changed by “prolonged run”.

The introduction should also describe and explain why healthy runners are an appropriate population to study for this investigation given that orthoses were originally designed to help patient populations. I suggest that the authors should discuss runner’s use of these products to prevent injury. The survey study in JOSPT on runner’s beliefs about injury may be helpful in this regard. Please also provide a rationale for examining knee flexion and be more explicit about why foot eversion was studied (I imagine that the reason is that orthoses are intended to modify eversion primarily, but whatever the reason should be stated).

Reply: We would like to thank the reviewer for this comment. We have modified the introduction following reviewer recommendation as follow (Page 3, Lines 46-53):

“The use of foot orthoses is one of the most used strategies in the treatment of overuse running injuries [3,4], being the most benefited runners those with an excessive pronation or a length discrepancy [5], or pathologies such as patellofemoral pain [6] or chronic Achilles tendon injury [7]. Despite the unclear evidence of the effect of foot orthoses in healthy recreational runners, its use is very usual in this population as a mechanism of injury prevention and performance improvement [8–10]. In addition, runners perceived that footwear, in which foot orthoses could be included, is one of the main factors associated with injury risk [11].”

On the other hand, we have added information on knee flexion and foot eversion, as requested (Page 3, Lines 62-65):

“The modification of those parameters could be associated with changes in kinematic parameters such as knee flexion and foot eversion [7,22], that are related with the injury incidence [23], which justifies its investigation.”

And (Page 4, Lines 70-73):

“By contrast, it has been speculated that foot orthoses could modify spatio-temporal kinematics [18] and angular kinematics during running, due to its usual aim of control of pronation [7,8].”

It did not appear that the authors examined the individual subject data. Footwear and foot orthoses research is notorious for finding insignificant differences, not because the conditions have no effect but because the inter-individual variation in the response to the conditions is so large.

Reply: We are totally agree with this comment about the inter-individual variation effects on the results. For this reason, we have carried out repeated-measures ANOVAs that compare each subject with himself in the different conditions, since each participant went through the 3 types of foot orthoses and the 5 time points of measurement. This type of analysis is taking into account the intra-individual variations as suggested by the reviewer. 

The discussion was a bit outside of the scope of the present study in some areas (for example, with respect to energy expenditure and what affect that may have on the results. The present study only examined HR and RPE, neither of which were between orthosis conditions). Any discussion regarding energy expenditure or running economy should be brief, to the point, and within the scope of the present study (i.e. one paragraph of no more than 10-12 lines in length). The scope of the paragraph discussing the knee flexion results (line 299-319) is a good example of the scope that should be used to explain the other variables and length should not exceed that of this example section, given the results and the overall scope of the study.

Reply: We are grateful to the reviewer for this comment. We agree with the reviewer that some of our discussion was outside of the scope of our study. Therefore, we had reduced the information of the discussion related with energy expenditure or running economy.

Similar to the introduction, the discussion section seems to focus on the differences between conditions in the primary outcome variables (knee angle, eversion angle, and spatio-temporal parameters), rather than the effects of the orthoses on the prolonged run. I understand that it is difficult to explain the effects that the orthoses had on the prolonged run given that there were no significant differences observed. However, the purpose of this suggestion is make sure the discussion surrounding the prolonged run aspect of this study is not seen as an afterthought; it should be the focus of both the intro and discussion because it is the primary difference compared with previous studies.

Reply: We would like to thank the reviewer for this comment. We modified our discussion increasing its focus in the prolonged run. We maintained the structure of discussing the primary outcome variables, because we think that the focus of prolonged run should be included in all the paragraphs of the discussion, and this structure is an easy way for the reader to follow the discussion of our results. 

Abstract

1. Line 21-23: I recommend revising this section of the abstract to reflect better why healthy runners would chose to wear foot orthoses or shoe inserts and thus why they are a good sample to study rather than a patient population.

Reply: We would like to thank to the reviewer for this comment. This section has been modified as follows (Page 2, Lines 21-23): 

“Foot orthoses are one of the most used strategies by healthy runners in injury prevention and performance improvement. However, their effect on running kinematics throughout the evolution of an intense prolonged run in this population is unknown.”

2. Line 36-38: I recommend changing the last sentence of the conclusion to be something that can be said about the present study. Additional work can always be done but a more impactful conclusion would reflect what can be learned by the present study. As discussed above, commenting about inter-individual differences may be a good topic to replace this last sentence (see further comment below about the inter-individual differences).

Reply: We would like to thank to the reviewer for this comment. We have changed the last sentence of the conclusion in order to focus more on the present study, as follow (Page 2, Lines 37-39):

“This lack of modifications suggest that a healthy runner population maintains its ideal movement pattern throughout a 20 minute prolonged run, regardless the type of foot orthosis used.”

Introduction

3. Line 45: Please be specific here as to not unintentionally spread speculation. The referenced study seems to be about Achilles tendon injuries alone. Also, please confirm that the findings of this study were indeed that the orthoses prevented injury, which is the statement in the text of the present study. Citations should reference the findings, not the speculation or theories proposed by others.

Reply: We would like to thank to the reviewer for this comment. This sentence has been changed to solve a previous comment, so this cite has been deleted because it does not support the new idea.

4. Line 60: “…no clear evidence…of the effect of foot orthoses on running kinematics” is not clear because the authors described some kinematic variables earlier in this paragraph. Please be more specific as to what is still unknown. Also in this paragraph, the ‘preferred movement path’ is discussed as a follow-up to describing that there is not a lot of evidence regarding spatio-temporal variables. Although both portions of the sentence are true individually, they should not be together because the preferred movement path is less a consideration for spatio-temporal characteristics, and more about joint and segment kinematics.

Reply: We would like to thank the reviewer for this comment. Different modifications were performed to address this comment. 

Firstly, the sentence “…no clear evidence…of the effect of foot orthoses on running kinematics” has been modified to be clearer (Page 4, Lines 81-82):

“In this sense, there is a lack of information regarding the kinematic effects of foot orthoses during a prolonged run in the literature.”

Secondly, the sentence about ‘preferred movement path’ has been connected with kinematic adjustments in general and with foot eversion, as follow (Page 4, Lines 70-81):

“By contrast, it has been speculated that foot orthoses could modify spatio-temporal kinematics [18] and angular kinematics during running, due to its usual aim of control of pronation [7,8]. The type of the foot orthosis could also have an influence in kinematic adjustments, expecting better results with custom-made foot orthoses than with prefabricated foot orthoses, due to its individual adaptation [28,29]. However, some authors have found a reduction in foot eversion with custom-made foot orthoses [9,14,30], while others have observed the same effect with prefabricated ones [17,31]. In this point, according to one of the last paradigms postulated by Nigg et al. [23,32], the use of footwear (or foot orthoses in this case) should allow healthy runners to maintain their preferred movement path, so caution should be taken with kinematic alterations, since they will not always be positive.”

Methods

5. The authors described that they tested if the variables were normally distributed but did not describe the results of these tests in the statistical analysis section of the methods or in the results. Please describe which variables, if any, were non-normally distributed and what was done (if anything) to adjust for non-normal data.

Reply: We would like to thank the reviewer for this comment. One sentence of the statistical analysis section has been specified to clarify it (Pages 8-9, Lines 189-191):

“After confirming the normality of the sample in all the variables by the Shapiro Wilks test (P > 0.05), and verifying the sphericity by the Mauchly Sphericity test, the descriptive statistics were extracted.”

Results

6. Line 215-216: The end line 216 which is part of the sentence, “there was no significant main effect…in any of the kinematic variables; however, there was a main effect for…” should be changed to “…in any of the kinematic variables except for a main effect for…”. More simply stated, please change “however” to “except for”

Reply: We would like to thank the reviewer for this comment. The sentence has been changed as proposed.

7. As described in the general/overall comments, it may be beneficial to examine the inter-individual differences.

Reply: We think that the reviewer want to reference the intra-individual and not the inter-individual differences. As discussed in a previous comment, our statistical analysis has examined intra-individual differences (comparing a subject with himself in different conditions).

Discussion

8. Line 250: The findings of the referenced study were not “harmful kinematic alterations.” We do not have enough longitudinal evidence to imply that any cross-sectional study is harmful.

Reply: We would like to thank the reviewer for this comment. The sentence has been modified to clarify, as follow (Page 11, Lines 255-257):

“It is well known that during the evolution of a high intensity run, and consequently, with the development of it, some kinematic alterations related to the likelihood of injury and the performance optimization can occur [24,27].”

9. Line 258: These are the only studies that looked at spatio-temportal parameters with respect to what? Orthoses? These are far from the only prolonged running studies out there. This sentence should be more specific.

Reply: We would like to thank the reviewer for this point. The sentence has been specified, as follow (Page 12, Lines 264-268):

“Only the studies of Lucas-Cuevas et al. [20,21] analysed some spatio-temporal parameters in a continuous running protocol, with three different foot orthosis conditions (similar to those employed in the present work), but only at the beginning and at the end of the run, without looking at their evolution.”

10. Line 275: Need to explain how the methods of referenced study (Wilkinson et al) are similar and different than the present study so the reader can have more context to understand why the results between the two studies are being compared and understand the potential explanation for the differences. Start the paragraph explaining this information rather than bring it in later (i.e. line 279-283).

Reply: We would like to thank the reviewer for this comment. We have modified the knee flexion paragraph, where this occurred, to solve a comment from the other reviewer, so we have removed this reference. However, we have taken this comment into account at other times in the discussion.

Figures and tables

11. There is a typo with the spelling of ‘length’ in Figure 4.

Reply: We would like to thank the reviewer for the correction. 

12. It is my personal preference to have the full variable name used as figure titles instead of the abbreviation. That way, I don’t have to remember or look back at what something is, I know it the instant that I look at it.

Reply: We would like to thank the reviewer for this comment. We have removed the abbreviations of the variables names, both in the text and in figures 5 and 6. In addition, we would like to comment that we have corrected the design of Figure 5, because we have found an error in the representation of the error bars.

Reviewer #2: Overall, the manuscript is well-written, with a few areas where the use of English grammar is not quite up to expectations. The authors do well to provide background review of the literature and justification for the study. It seems the gap that the authors hope to fill is the effect of custom orthoses over time-course of an intense continuous run. However, the findings of the study indicate that there was no statistical difference to report. A null finding is not considered a negative thing, but it does substantially reduce the enthusiasm for this study. Additionally, while the authors have provided a thorough treatment of the data and kept their conclusions within the scope of the statistical results, they do tend to overstate potential application of this work in the Conclusion statement. I'm not sure that enough evidence was presented to say that impact reduction with orthoses can be a benefit "without interfering in the running technique or running economy".

Reply: We would like to thank the reviewer for the comment. We have changed the last sentence of the conclusion, as follow (Page 16, Lines 376-378):

“This lack of modifications suggest that a healthy runner population maintains its ideal movement pattern throughout a 20 minute prolonged run, regardless the type of foot orthosis used.”

There are a large number of figures, which are unfortunately not very informative. Figures 1-3 are generally helpful to explain the methods, but the information in Figures 4-6 could easily be placed in a table, or in supplementary material, as they really do not show anything more than what the authors have stated in the text.

Reply: We would like to thank the reviewer for this suggestion, but we consider interesting that the figures 4-6 (data of the variables) appear in the final version in order to contextualize the results obtained. However, if the reviewer considers it necessary, we could add them as supplementary material. We had checked this data in a table and we concluded that the figures will be more comprehensible for readers. 

The discussion section is excessively long, with too much revisitation of the previous work. Rather than re-reviewing all of the literature here, the manuscript would be improved with a critical evaluation of the points in the literature which may help explain why the current study resulted in null findings.

Reply: We would like to thank the reviewer for this comment. The review of the literature has been reduced in some cases, and it has tried to improve the explanation of the results.

---

## [Decision Letter · Decision Letter 1]

10 Feb 2020

PONE-D-19-28910R1

Effect of custom-made and prefabricated foot orthoses on kinematic parameters during an intense prolonged run

PLOS ONE

Dear Dr. Jimenez-Perez,

Thank you for submitting your manuscript to PLOS ONE. After careful consideration, we feel that it has merit but does not fully meet PLOS ONE’s publication criteria as it currently stands. Therefore, we invite you to submit a revised version of the manuscript that addresses the points raised during the review process.

In your revision, please consider revising Figs. 4-6 and/or compile them into tables for clarity, correct the grammatical errors in the manuscript, and discuss and present individual data as well as if there were responders and non-responders.

We would appreciate receiving your revised manuscript within 60 days. To enhance the reproducibility of your results, we recommend that if applicable you deposit your laboratory protocols in protocols.io, where a protocol can be assigned its own identifier (DOI) such that it can be cited independently in the future. For instructions see: http://journals.plos.org/plosone/s/submission-guidelines#loc-laboratory-protocols

We look forward to receiving your revised manuscript.

Kind regards,

Alena Grabowski

Academic Editor

PLOS ONE

Reviewers' comments:

Reviewer's Responses to Questions

**Comments to the Author**

1. If the authors have adequately addressed your comments raised in a previous round of review and you feel that this manuscript is now acceptable for publication, you may indicate that here to bypass the “Comments to the Author” section, enter your conflict of interest statement in the “Confidential to Editor” section, and submit your "Accept" recommendation.

Reviewer #1: (No Response)

Reviewer #2: (No Response)

2. Is the manuscript technically sound, and do the data support the conclusions?

Reviewer #1: Yes

Reviewer #2: Yes

3. Has the statistical analysis been performed appropriately and rigorously? 

Reviewer #1: Yes

Reviewer #2: Yes

4. Have the authors made all data underlying the findings in their manuscript fully available?

Reviewer #1: Yes

Reviewer #2: Yes

5. Is the manuscript presented in an intelligible fashion and written in standard English?

Reviewer #1: Yes

Reviewer #2: Yes

6. Review Comments to the Author

Reviewer #1: The introduction and discussion are much improved now with a more narrow focus that is within the scope of the study.

Many grammatical errors remain that should be addressed.

The purpose of my previous comment regarding the individual differences was to emphasize that each individual will have a very different response to a given footwear or orthotic condition. I understand the use of repeated measures statistics; however, my comment was a recommendation to show the individual data in the figures and tables (as supplements would be fine) instead of the group means and standard deviations as well as to suggest some discussion regarding if there were responders and non-responders, for example. Revealing that some subjects had more of a response than others may result in an alternative perspective or interpretation of the results than what is currently described in the discussion section.

Reviewer #2: The authors have addressed most of my previous concerns. I still feel that the data presented in Figures 4-6 would be better presented in a table form. With essentially equal means and very high deviation bars, the graphs really don't reveal anything more insightful than what could be shown in a table.

7. PLOS authors have the option to publish the peer review history of their article (what does this mean?). If published, this will include your full peer review and any attached files.

Reviewer #1: No

Reviewer #2: No

---

## [Author Response · Author response to Decision Letter 1]

24 Feb 2020

Thank you for submitting your manuscript to PLOS ONE. After careful consideration, we feel that it has merit but does not fully meet PLOS ONE’s publication criteria as it currently stands. Therefore, we invite you to submit a revised version of the manuscript that addresses the points raised during the review process.

In your revision, please consider revising Figs. 4-6 and/or compile them into tables for clarity, correct the grammatical errors in the manuscript, and discuss and present individual data as well as if there were responders and non-responders.

Dear editor, we are grateful for the opportunity to revise our paper. All the comments sent by the reviewers were carefully addressed. All the changes in the edited manuscript are highlighted with red text in the revised submission. We look forward to hearing from your editorial decision.

Reviewers’ comments:

Reviewer #1: The introduction and discussion are much improved now with a more narrow focus that is within the scope of the study.

Reply: We are grateful to the reviewer for this comment.

Many grammatical errors remain that should be addressed.

Reply: The manuscript has been carefully reviewed to correct all grammatical errors.

The purpose of my previous comment regarding the individual differences was to emphasize that each individual will have a very different response to a given footwear or orthotic condition. I understand the use of repeated measures statistics; however, my comment was a recommendation to show the individual data in the figures and tables (as supplements would be fine) instead of the group means and standard deviations as well as to suggest some discussion regarding if there were responders and non-responders, for example. Revealing that some subjects had more of a response than others may result in an alternative perspective or interpretation of the results than what is currently described in the discussion section.

Reply: We would like to thank the reviewer for this comment. Three figures with the percentage of change to each biomechanical variable for each participant, and one table that presents the number of participants that showed a biomechanically relevant change have been added as supplementary material. This is mentioned in the results section (Page 13, Lines 252-254):

“Percent changes of the biomechanical variables for each participant as well as the number of participants that showed a biomechanically relevant change (±10%) [36] regarding the control condition are provided in the Supplementary File.”

In addition, at the end of the discussion, a paragraph has been added regarding individual differences, as suggested (Page 18, Lines 375-385):

“From this point of view, although no differences have been found between the three foot orthosis conditions studied from a grouped perspective, in the studies of foot orthosis and footwear it is common to find a large variability of response among participants, which is masked by the overall results [36]. In the present study, as presented in the Supplementary File, the spatio-temporal parameters have not been affected by inter-subject variability, since most participants have not experienced a relevant change. However, in the angular parameters, the effects of different directions and magnitudes have been observed both among participants as well as within a single individual. In the future, it would be interesting to study if these different responses to a foot orthosis implementation could be grouped by some participant characteristics, such as the gender or the type of foot.”

Reviewer #2: The authors have addressed most of my previous concerns. I still feel that the data presented in Figures 4-6 would be better presented in a table form. With essentially equal means and very high deviation bars, the graphs really don't reveal anything more insightful than what could be shown in a table.

Reply: We would like to thank the reviewer for this comment. The data showed in Figures 4-6 have been presented in tables (Tables 2, 3 and 4, respectively) as suggested.

---

## [Decision Letter · Decision Letter 2]

11 Mar 2020

Effect of custom-made and prefabricated foot orthoses on kinematic parameters during an intense prolonged run

PONE-D-19-28910R2

Dear Dr. Jimenez-Perez,

We are pleased to inform you that your manuscript has been judged scientifically suitable for publication and will be formally accepted for publication once it complies with all outstanding technical requirements.

With kind regards,

Alena Grabowski

Academic Editor

PLOS ONE

Additional Editor Comments (optional):

Reviewers' comments:

Reviewer's Responses to Questions

**Comments to the Author**

1. If the authors have adequately addressed your comments raised in a previous round of review and you feel that this manuscript is now acceptable for publication, you may indicate that here to bypass the “Comments to the Author” section, enter your conflict of interest statement in the “Confidential to Editor” section, and submit your "Accept" recommendation.

Reviewer #1: All comments have been addressed

Reviewer #2: All comments have been addressed

2. Is the manuscript technically sound, and do the data support the conclusions?

Reviewer #1: Yes

Reviewer #2: Yes

3. Has the statistical analysis been performed appropriately and rigorously? 

Reviewer #1: Yes

Reviewer #2: Yes

4. Have the authors made all data underlying the findings in their manuscript fully available?

Reviewer #1: Yes

Reviewer #2: Yes

5. Is the manuscript presented in an intelligible fashion and written in standard English?

Reviewer #1: Yes

Reviewer #2: Yes

6. Review Comments to the Author

Reviewer #1: (No Response)

Reviewer #2: The authors have adequately addressed my previous concerns. The revised version of the manuscript now reads very clearly and the presentation of results is concise and clear.

7. PLOS authors have the option to publish the peer review history of their article (what does this mean?). If published, this will include your full peer review and any attached files.

Reviewer #1: No

Reviewer #2: No

---

## [Editor Report · Acceptance letter]

13 Mar 2020

PONE-D-19-28910R2 

Effect of custom-made and prefabricated foot orthoses on kinematic parameters during an intense prolonged run 

Dear Dr. Jimenez-Perez:

I am pleased to inform you that your manuscript has been deemed suitable for publication in PLOS ONE. Congratulations! Your manuscript is now with our production department. 

With kind regards,

on behalf of

Dr. Alena Grabowski 

Academic Editor

PLOS ONE